# Learning from fights: Males' social dominance status impact reproductive success in *Drosophila melanogaster*

Antoine Prunier, Severine Trannoy [ID] *

Research Center on Animal Cognition (CRCA), Center for Integrative Biology, Toulouse University, CNRS, UPS, Toulouse, France

* severine.trannoy@univ-tlse3.fr

## Abstract

In animals, the access to vital resources often relies on individuals' behavioural personality, strength, motivation, past experiences and dominance status. Dominant individuals would be more territorial, providing them with a better access to food resources and mate. The so-called winner and loser effects induce individuals' behavioural changes after experiencing a victory or a defeat, and lead to an individual persistent state influencing the outcome of subsequent fights. However, whether and how development of winner and loser effects affect individuals' fitness is controversial. The aim of this study is to evaluate how individuals' fitness can be influenced by previous fighting experience in *Drosophila melanogaster*. In this study, we assess various behavioural performances as indicators for dominant and subordinate fitness. Our results show that subordinates are less territorial than dominants although their locomotor abilities are not affected. We also demonstrate that in a non-competitive context, experiencing a defeat reduces males' motivation to court females but not the reproductive success while in a competitive context, it negatively affects males' reproductive success. However, we found no impact upon either males' ability to distinguish potential mates nor on females' choice of a specific mating partner. Overall, these results indicate that previous defeats reduce reproductive success, a commonly used estimate of individual fitness.

## Introduction

One of the fundamental characteristics of animals is their ability to react appropriately in a constantly changing and challenging environment, which ultimately benefits their fitness *i.e.* the probability of an individual to survive and its capacity to pass its genetic material to its offspring [1]. Behavioural adaptation relays on crucial processes such as integration of environmental sensory cues and animals' past experiences. Indeed, animals can perceive relevant information such as the presence of conspecifics or food resources and generate appropriate behavioural responses, while their past experiences are used to provide adaptive responses to changing environmental conditions and provide behavioural flexibility.

**Data Availability Statement:** All relevant data are within the manuscript and its Supporting Information files.

**Funding:** This work was supported by the French Research National Agency (ANR) (ANR-19-CE37-

0018-01), and the Fondation Fyssen (190573) to S. T. Funders had no role in study design, data collection and analysis, decision to publish or preparation of the manuscript. https://anr.fr/ https://www.fondationfyssen.fr/en/.

**Competing interests:** The authors have declared that no competing interests exist.

Animals often adjust their behavioural responses to information provided by social interactions. Among the latter, aggressive behaviours are widespread across the animal kingdom. They are essential to the establishment of group structure, and to access to primary resources (*e.g.* food, territory, sexual partners. . .) or protection [2]. For this reason, aggression has a high adaptative value and is often observed in the context of inter-specific competition. On one hand, animals need to assess the benefits/risks balance when engaging into aggressive interactions: is there something to fight for and is it something valuable? Against whom? Is it risky? [3]. On the other hand, animals adapt the pattern and intensity of their aggressive behaviours as well as their fighting strategy depending on the outcomes of previous fights. Such behavioural adaptation is highlighted by the formation of winner and loser effects, consisting in the increasing probability to win subsequent fights after victories, or to lose after defeats, respectively [4].

In this study, we used the fruit fly *Drosophila melanogaster* as an animal model capable of various social behaviours: aggregation [5], courtship [6] and intra-sex aggression [7]. In fruit flies, cuticular hydrocarbons (CHC), such as some pheromones—*e.g.*: (Z)-7-tricosene (7T) or 11-cis-vaccenyl acetate (cVA)—play major roles in individuals' identification, and drive behavioural responses towards courtship or aggression [3, 8–10]. When competing for food or mates, male flies start interacting with each other until one decides to display some physical attacks, called lunges. In general, after multiple lunges from the same male, the second one retreats from the food resource. Wing threats can also be observed as offensive or defensive behaviours. In rare cases, high-intensity aggressive pattern can be observed, such as tussling or boxing [7]. The repetition of this behavioural sequence—lunges/retreats—allows the formation of dominance relationship between competitors, that will generate winner and loser states, involving learning and memory processes [11, 12].

Winner and loser effects are found in a wide range of species including humans [13], mammals [14], fish [15], birds [16], crustaceans [17], arthropods [18] and in fruit flies [11], and may durably impact animals' behaviour. Contests may influence behaviour in various ways according to the sex and the species. In males, these effects can be explained by an increased motivation to fight after previous victories while it is reduced following defeats, with a central role given to sexual steroids such as androgens and monoamines [19, 20]. For instance, androgens level is increased in cichlid fish when fighting, even towards its own reflection, and increases the likelihood of winning subsequent encounters [15], while in rodents, winning fights induces testosterone pulses [20]. Another study found an increased tyrosine hydroxylase expression level in winners' brain regions controlling social and motivational behaviours [19]. In *D. melanogaster*, single victory and defeat respectively increases and decreases the likelihood to win and lose subsequent fight for few minutes to hours [11]. Recently, it has been shown that serotonin is involved in the short-term loser effect [21]. However, repeated defeats induce long-lasting loser effects for at least a day and is dependent on *de novo* protein synthesis [11, 22]. A remarkable case was observed in the mangrove killifish, where a previous defeat continues to exert its influence on the outcome of further fights, even a month later. These show that previous fighting experiences may profoundly affect individuals, from their behaviour to their genes and physiology, in both short- and long-term ways.

While previous victory confers fighting advantages to winning individuals, the extent to which it may also confer mating advantages and may potentially increase males' overall fitness remains a subject of controversy. In the mosquitofish for instance, victories increase reproductive success [23]. In males jumping spider *Thiania bhamoensis*, quivering courtship behaviour is enhanced after winning, which increases reproductive success, even though the females tend to prefer loser males when she did not watch the fight [24]. In crickets *Gryllus assimilis* however, females have a preference for dominant males, whether they saw the fight or not [25].

Studies on *D. melanogaster* and *prolongata* have revealed that subordinated status is associated with longer latency to court and less successful reproductive behaviour while dominant status do not improve reproductive success compared to naïve flies [22, 26–29]. However, the reproductive males' advantages seemingly provided by the winner effect is not observed in all species. In the earwig *Euborellia brunneri*, only loser effect has been observed in males, and females do not show preference towards winner, loser or naïve males [30]. In the olive fruit fly *Bactrocera oleae*, both winners and losers showed higher aggressive behaviours against naïve flies. [31]. In the fruit fly, it has been found that winners get more access to the female but the losers showed higher post-copulatory investment [29]. Hence, winner and loser effects influence the outcome of further social interactions either with potential same-sex competitors or mates, and represent *in fine* one of the mechanisms contributing to natural selection. Nonetheless, the consequences and mechanisms of how previous fighting experience may impact reproductive success still remain to be clarified.

The aim of the present study was to explore whether and how previous fighting experiences influence some behavioural aspects essential for males' reproductive success. We first assessed the territoriality of males and present evidence that subordinate males are less territorial than dominants, despite that their locomotor abilities are not affected, leading to a potential disadvantage to access food resources or potential mates. Then, we examined the motivation to court females, and show that previous defeat alters this motivation to court females without affecting the reproductive success of subordinates. However, in a competitive context, subordinates presented a reduced reproductive success over dominant males. Finally, we demonstrate that males' perception is not modified by past fighting experiences, and that females did not present preferences for males with dominant social status. Overall, this study demonstrates the impact of males' social dominance status on their future behaviours and further social interactions with conspecifics, ultimately leading to potential impact on their fitness.

## Materials and methods

### Flies

The *Drosophila melanogaster* from the strain *Canton-Special* (CS) was used for this study. Flies were raised on standard food medium at 25°C and 50% relative humidity in incubators under a 12:12 hours light:dark cycle with white LED strips.

For behavioural tests, males and females were reared in social isolation from late pupal stage to the day of testing to ensure reliable aggression [32]. They were kept in 5ml vials closed by a piece of cotton and containing about 1ml of standard fly food. Behavioural assays were performed at 7-days old, between 9:00 and 12:00 which is the first 3 hours of the daily light cycle.

### Experimental chamber

The experimental setup used in this study to examine social behaviours has already been described [33]. Briefly, tests were performed in a circular arena (22mm diameter x 16mm height) with a food cup in the middle. An opaque plastic divider was inserted from the top of the lid, separating the arena in two equal sides. Flies were then inserted on each side of the arenas by negative geotaxis, and had 10min to acclimatise, without interacting with each other. Behavioural experiments started once the separator was removed, allowing flies to interact together. Each experiment was performed at 25°C.

## Behavioural assays

**Aggression assays and establishment of winner or loser state.** The experimental protocol was previously described in Trannoy, et al. [33]. Flies were anesthetized with $CO_2$ 48 hours before the behavioural assays and a dot of white acrylic paint was applied on half of the tested flies on the dorsal thorax to facilitate visual tracking of individual flies. The other half was anesthetized with $CO_2$ but were not painted. A cup (13x6mm) filled with standard food medium (food cup) and with a drop of yeast paste in the middle were used as an attractive resource in all fights, and was placed in the middle of the arena and fixed with adhesive paste. The arena was separated with a divider so both inserted flies could have access to half of the resource without being in physical contact with the other male. Subordinate males were defined as males that retreated at least 3 times from the food cup after receiving lunges from the opponent. After 20min of fight, the divider was inserted to separate dominants and subordinates. It was important to not stress flies by inserting the divider nor induce flight otherwise results might be biased. Then, after having being separated for 10 or 60 minutes after fighting experiences, dominants and subordinates were tested for their territoriality, locomotion, courtship and preferences assays described below. Males were tested only once for behavioural assays and one timing (10 or 60 minutes).

**Territoriality assays.** Directly after the 20min fights, an opaque divider was inserted within the arena to separate dominants and subordinates. Then, we recorded their territoriality on each side of the food cup during 5min. Territoriality of dominant and subordinate fly were scored in BORIS v.7.12.2 [34]. The time to return to the food cup and the time spent on it have been measured for 5min, both starting from the insertion of the divider. Some flies were already on the cup when we inserted the divider, we thus excluded these flies from the measure "Time to return to the food cup" on Fig 1B. The fly was considered on the food cup if at least three legs of the fly touched the cup or the food, edges of the cup included. It was then expressed in percentage of time spent on the territory.

**Locomotion.** After a period of 10 or 60min once the 20min fight has ended, dominants and subordinates were inserted alone in a novel arena without food resource and were allowed to acclimate for 1min. Then, the number of midlines crossing was counted during 5 minutes.

**Courtship assays.** After a period of 10 or 60min once the 20min fight has ended, dominants and subordinates were separated from each other by inserting a divider. Then,

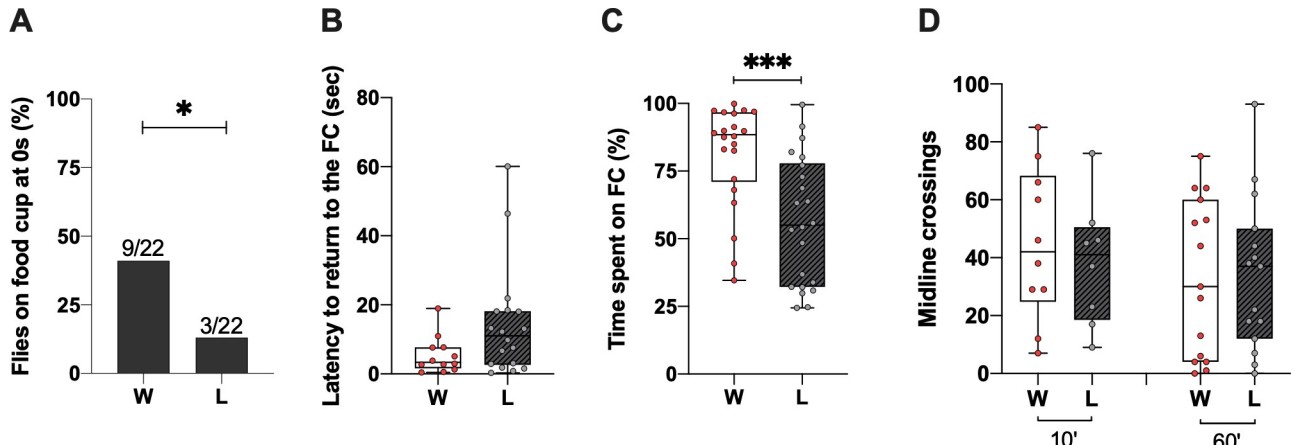

**Fig 1. Previous defeat had negative impact on males' territoriality.** (**A**) The proportion of flies starting the territoriality assays on the food cup was significantly larger in winners than losers (n = 22). (**B**) The time to return to the food cup was not significantly different between winners and losers (n = 22). (**C**) The proportion of time spent on the food cup was increased in winners compared to losers (n = 22). (**D**). The number of midline crossings 10 and 60min after fights was not different between losers and winners (10min: n>8; 60min: n = 15). N = Naïve, W = Winner and L = Loser. *$p<0.05$ **$p<0.01$; ***$p<0.001$.

dominants, subordinates, and naïve males (males that were inserted within the same arena used for aggression assays, but alone) were transferred into one half of a new arena without food resource, but containing a living virgin CS female on the other half. Courtship Vigour Index (CVI) was calculated as $\frac{T_t + T_{UWE} + T_{Ch} + T_{ac}}{T_{Obs}}$ where $T_t$ is the time the male did tapping, $T_{UWE}$ the time of Unilateral Wing Extension (UWE), $T_{Ch}$ the time of chasing, $T_{ac}$ the time of attempted copulation and $T_{Obs}$ the total time of the observation which was of 10min after the first courtship behaviour. The latencies to court and to mate were the times between the first meeting (*i.e.* the first physical contact between both flies) and the first courtship behaviour or initiation of mating. The number of abdomen binding were counted and the reproductive success (*i.e.* mating before 10min) for each male was scored.

**Male preference assays.** After a period of 60min once the 20min fight has ended, dominant and subordinate males were inserted into a novel arena containing a food cup. On the food cup, a decapitated virgin CS female and a decapitated CS naïve male were placed. To decapitate Cs females and males, they were anesthetized within a clean cold vial for 30 seconds. Using a clean razor blade, we cut off the heads of the sleepy flies. Their body were then put on each side the food cup with clean forceps. Decapitation was used to score interacting time of the test fly itself without bias due to behaviours of target flies. The time spent to interact with each were quantified. The amount of time obtained allowed the calculation of an index representing the time spent with a preferred individual. A positive index of 1 would reflect that a given male spent all the observation time interacting with the immobilized female. Conversely, a negative one of -1 would reflect that the male spent all the observation time interacting with the immobilized and naïve male. The index was calculated as equal to $\frac{T_F - T_M}{T_{Obs}}$ where $T_F$ is the total time spent with the female, $T_M$ with the male and $T_{Obs}$ the total time of the observation, everything being expressed in seconds.

**Female preference assays.** After a period of 60min once the 20min fight has ended, dominant and subordinate males were used for preference assays. In a new arena containing a food cup, one virgin CS female was inserted on one half of it. On the food cup, a decapitated dominant and subordinate were placed, using the same decapitation protocol described above. Once the divider was removed, the female was allowed to interact with immobilized males. As in the male preference assays, the time spent interacting with the dominant or the subordinate male allowed the calculation of an index reflecting the proportion of time and the preferred male, from -1 to 1. A positive index of 1 would reflect that female spent all the observation time interacting with dominants, while a negative one of -1 would reflect that they spent all her time interacting with subordinates. The index was calculated as equal to $\frac{T_W - T_L}{T_{Obs}}$ where $T_W$ is the total time spent with the winner, $T_L$ with the loser and $T_{Obs}$ the total time of the observation, everything being expressed in seconds.

**Courtship competition assays.** Males fought during 20min to establish dominance relationship. They were then separated with the divider and kept in the same arena. After 60min, a virgin female was inserted in arenas which met dominance criteria, randomly from the side of the dominant or subordinate. Then, the separator was immediately removed. Males were allowed to compete for the female until a male succeeded to mate with the female. The status of the first male displaying the first courtship event (*i.e.* one-wing extension), and the one which succeeded to copulate with the female were noted.

## Statistical analysis

Statistical analysis was performed using GraphPad Prism 8.2.1 (GraphPad Software, San Diego, California USA) and RStudio 2021.09.1+372. All data were subjected to a Grubb's test

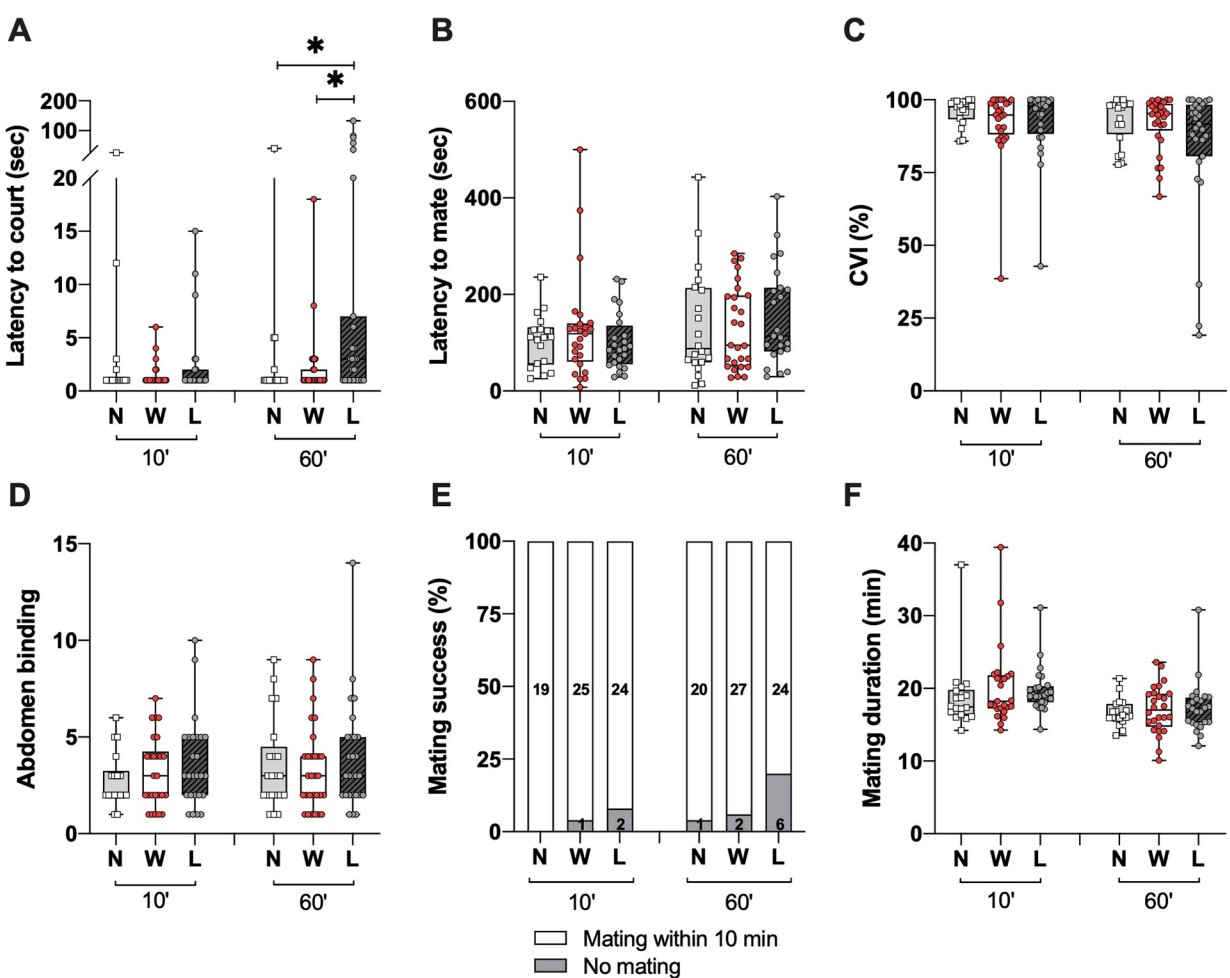

**Fig 2. Losing fight influence males' motivation to court females.** (**A**) The latency to court virgin CS females was significantly increased in losing males. However, the latency to mate (**B**), the Courtship Vigor Index (CVI) (**C**), the number of abdomen binding (AB) (**D**), the mating success (**E**) and the mating duration (**F**) were not significantly affected. (10min: $n_N = 19$ $n_W = 26$ $n_L = 26$; 60min: $n_N = 21$ $n_W = 29$ $n_L = 30$). *p<0.05; ***p<0.001.

($\alpha = 0.05$) to determine whether extreme values were outliers and were excluded from analysis. Since most of the datasets did not pass a Shapiro-Wilk normality test, non-parametric statistical tests were used for all our data analysis. Mann-Whitney and Kruskal-Wallis tests were used to compare data. Dunn's multiple comparisons test was used as a post-hoc test to determine which groups significantly differed. Survival analysis for CVI data (S1 Fig) were performed with RStudio by using survival and survminer packages and were compared using the log-rank test. Proportions (Figs 1A, 2E and 3A) were compared on RStudio by using Generalized Linear Model (GLM) following a binomial distribution and analysed with a Type-II ANOVA.

## Results

### Dominance status and territoriality

As a first step to investigate the influence of males' dominance status, we have assessed males' territoriality directly after 20min-fights. Following the separation of both opponents within the arena, we noticed a higher proportion of dominants already on the food cup when starting the assays compared to the losers (Chi$^2$ = 4.2712, df = 1, p = 0.0388) (Fig 1A). For flies that

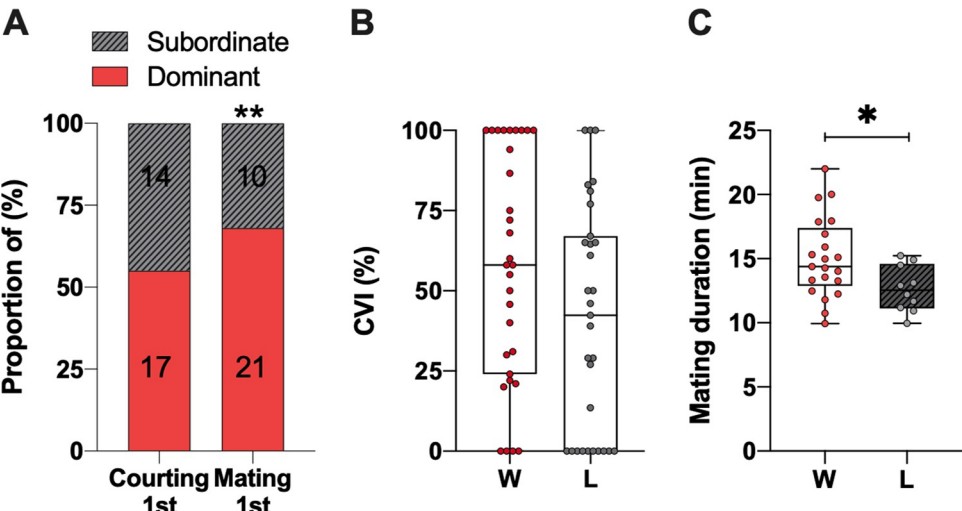

**Fig 3. Previous losers were less likely to mate upon winners.** (**A**) The likelihood to courting first the female was equivalent between winners and losers, however, previous winners had increased mating success (17 dominants did the first courtship behaviour versus 14 subordinates, and 21 dominants mated first versus 10 subordinates). (**B**) The CVI from dominants and subordinates were not statistically different from each other. (**C**) The mating duration was significantly longer for winners compared to losers *p<0.05 (n = 31).

were not on the food cup at the start of the assays, we observed no significant difference in the time to return to the food cup between subordinates and dominants (Mann-Whitney test, U = 69, p = 0.1038) (Fig 1B). However, dominants spent significantly more time on the food cup than the subordinates during the 5min of observation (Mann-Whitney test, U = 92, p = 0,0003) (Fig 1C). The difference cannot be assigned to locomotor deficit as any statistical differences between winners and losers were observed when measured 10min or 60min after fights (Mann-Whitney test, 10min U = 34.5, p = 0.6486; 60min U = 112, p = 0.9919) (Fig 1D). These results showed that winning a fight enhances males' attraction to the food cup and potentially their territoriality.

## Dominance status and courtship behaviour

Beside territoriality, the capacity to meet and court females to ultimately mate and produce off-spring is crucial for animals' survival. To assess the influence of previous fighting experience on courtship and mating success, we performed male-male aggression assays followed by courtship assays by pairing independently previous winners and losers with a CS virgin female. The results revealed that previous losers showed a significant increase in their latency to court females compared to winner and naive males 60min after fights (Kruskal-Wallis test, H = 10.68, p = 0.0048 and post-hoc Dunn's multiple comparisons test $p_{Naive>Losers}$ = 0.0143; $p_{Winners>Losers}$ = 0.0164 and $p_{Naive>Winners}$>0.999), but not after 10min (Kruskal-Wallis test, H = 0.5570, p = 0.7569) (Fig 2A). However, the latency to mate did not differ after both timings (Kruskal-Wallis test10min, H = 0.6380 and p = 0.7269; Kruskal-Wallis tests60min, H = 1.177, p = 0.5552) (Fig 2B). Similarly, the CVI which reflects the time spent by males courting females for 10min after the first courtship event, and the number of abdomen binding (AB) were not significantly different in their intensity (CVI: H10min = 0.5680, p = 0.7528; H60min = 2.206, p = 0.3319; AB: H10min = 1.421, p = 0.4914; H60min = 0.6318, p = 0.7291) (Fig 2C and 2D) nor in their extinction (10mn: $Chi^2$ = 0.4, Df = 2 and p = 0.83; 60mn: $Chi^2$ = 3.5, Df = 2 and p = 0.17) (S1A and S1B Fig). Then, to assess mating behaviours, we analysed the mating

success and the persistence to mate. Both mating success and mating duration were not significantly different after both timings (mating success: $Chi^2 10min$ = 2.1898, df = 2, p = 0.3346; $Chi^2 60min$ = 3.6535, df = 2, p = 0.1609; Mating duration: Kruskal-Wallis test10min, H = 4.053, p = 0.1318; H60min = 0.4939, p = 0.7812) (Fig 2E and 2F). These results demonstrate that only previous losing experiences negatively affect males' motivation to court females 60 min after fights, but had no significant influences on other courtship and mating capacities.

### Dominance status and reproductive success in a competitive context

Dominant and subordinate males did not present drastic differences in their abilities to court and mate, but a decreased in the subordinates' motivation to court females 60 minutes after fights has been observed. Thus, we then asked whether dominance status would have an impact on reproductive success when the pressure of competition is strong. To tackle this, 60 minutes after fights, we inserted a CS virgin female within the arena and scored courtship behaviours from both males. In such context, we demonstrated that winners and losers had the same probability to perform the first courtship event ($Chi^2$ = 0.5816, Df = 1, p = 0.4457), whereas losers had a significantly lower probability to mate first compared to winner males ($Chi^2$ = 7.9791, Df = 1, p = 0.0047) (Fig 3A). Subordinates' courtship performances were not quantitatively altered as their CVI did not differ significantly from that of the dominants, even though a trend can be noticed (Mann-Whitney test, U = 350,5, p = 0.0649) (Fig 3B). However, over time, the losers' CVI shows a faster decline compared to that of the winners ($Chi^2$ = 4.3, df = 1, p = 0.037) (S1C Fig). This suggests that subordinates may displayed fewer UWE explaining their lower probability to mate. Yet, when they succeed, we found that the mating duration differed significantly, such that the winners had longer mating duration than the losers (Mann-Whitney test, U = 55, p = 0.0348) (Fig 3C). Our results showed that previous losing experiences have mostly negative consequences on reproductive success only when individuals face competitors.

### Males and females' preferences

To explain the lower reproductive success of subordinates in a competition context, we carried out preferences assays. First, a previous defeat may lead to an impairment of correctly identifying females or a reduction of motivation to interact, court and ultimately mate with females. To disentangle these points, 60 minutes after fights, previous winners and losers had the choice to interact with a decapitated CS virgin female or CS naïve male. Both winner and loser males showed a clear preference for interacting with the decapitated female, and the time of interaction did not differ between both males (Mann-Whitney test, U = 6, p = 0.4127). They also showed the same lack of interest for the decapitated naive male (Mann-Whitney test, U = 6, p = 0.4127) (Fig 4A). As a consequence, the male preference index did not differ significantly between winners and losers (Mann-Whitney test, U = 7, p = 0.5635; $m_{win}$ = 0.91, $sd_{win}$ = 0.12 and $m_{los}$ = 0.86, $sd_{los}$ = 0.14) (Fig 4B). Second, females may be more receptive to dominant males' status, and would be more willing to mate with previous winners. To test this hypothesis, CS females were able to interact with decapitated winners and losers that have fought 60 minutes before, and we measured the time of interaction with both decapitated males. The results showed that females spend as much time interacting with winners as losers (Mann-Whitney test, U = 203, p = 0.3664) (Fig 4C), which is also outlined by a females' preference index around 0 (m = -0.08, sd = 0.43) (Fig 4D). These results suggest that previous defeat may directly impact males' motivation to court, and to the end, affect their reproductive success in a competitive context, rather than on females' preference for dominant males.

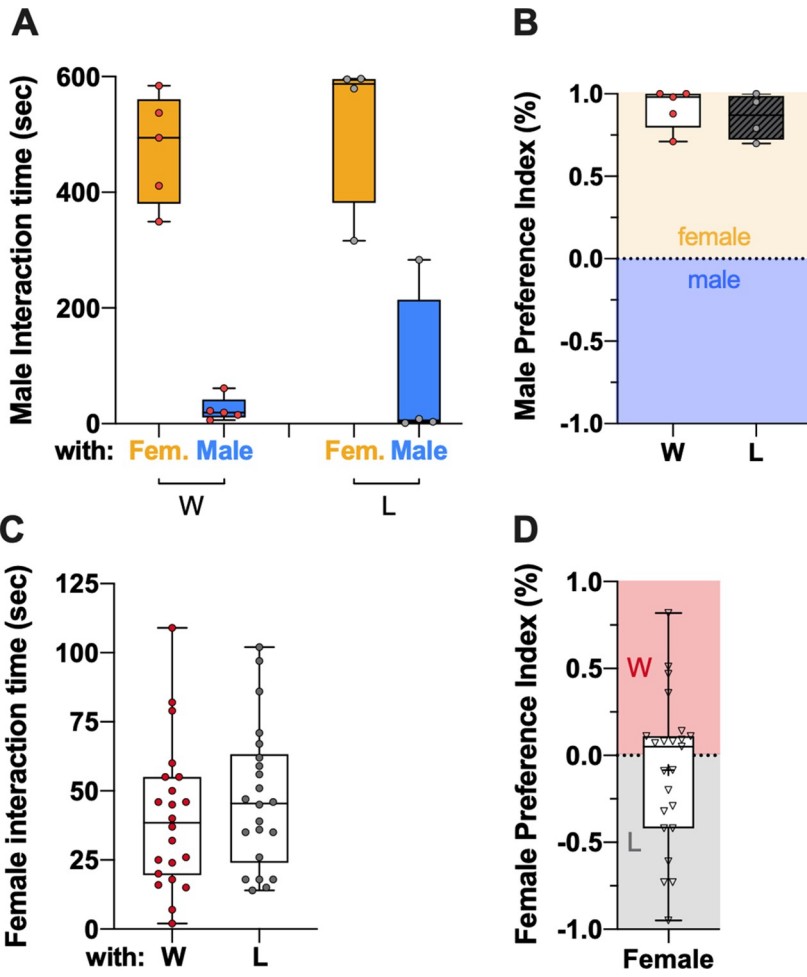

**Fig 4. Previous fighting experiences did not alter males' perception for females.** (**A**) Previous losers spend as much time than winners interacting with decapitated females. They spend more time interacting with females than males. (**B**) Male preference Index was equivalent between winners and losers (n>4). (**C**) CS females spend as much as time interacting with decapitated winners and losers (**D**) Female preference Index is equivalent to zero, showing no preference for winner or loser males (n = 23).

## Discussion

The aim of our study was to determine the short-term effects of fights' outcome on some *Drosophila melanogaster* males' behavioural aspects that are considered as proxies for individual fitness. First, we examined the territoriality with respect to the contested resource when dominants and subordinates were physically separated directly after fighting experiences. 40% (9/22) of winners, but only 13% (3/22) of losers, were still present on the food cup at the beginning of the assays, meaning that they did not leave the food cup between the end of the fight and the start of the territoriality assays. This difference suggests that winners may demonstrate greater territoriality or motivation to access food compared to losers. When assessing the time it took for flies that were not initially on the food cup to return to it, no significant difference was observed between the two groups. However, defeated males tend to take a longer period of time to return to the food cup compared to victorious males. Besides, despite the absence of physical contact with dominant flies during the territoriality assay, defeated males presented a reduced time spent on the food cup. Access and defence of a territory has been shown to be an

advantage for males' reproductive success since females aggregate on it [35]. Therefore, the reduced territoriality behaviour of subordinates may directly lead to decrease likelihood to mate. Such results demonstrate that fights have rapid effects on individuals' behaviours that persisted even in absence of physical contact with their opponent. These observations were likely due to changes in internal state of flies induced by fighting experiences, as winning or losing fights did not impair males' locomotor activities nor sex-identification. Previous studies have shown that fighting strategy adopted by dominant and subordinate males is influenced by potential territory markings [36], implying that the social-cue theory would explain behavioural changes after fighting experiences. This theory entails that individuals are able to perceive environmental sensory cues deposited by the opponent to guide further behaviours and social interactions [37]. To test whether the social-cue theory would explain why subordinate males were less territorial because they could detect territorial markings left by dominants during fights, one could perform the same territoriality assays but in novel arena exempt of any males' pheromones cues.

We then assessed the courtship and reproductive behaviours of dominant, subordinate and naïve males at two different timings after a fighting experience, 10 and 60 minutes. Overall, no behavioural differences were statistically detected between the three groups of males 10 mins after fights. However, 60 mins after previous defeat, subordinates present a decreased in their motivation to court virgin females, and tend to display fewer UWE. This shows that previous defeat can affect males' behaviours at different timing scales, inducing immediate changes in term of territoriality, but short-term changes in term of motivational internal state that develop 60min after fights. However, courtship behaviours 10min after fights are not affected in losers. This could be explained by the fact that winning and losing experiences, as any social interactions and environmental changes, may induce specific quantitative changes in the cuticular profiles of winners and losers [38–40]. Such changes would require new synthesis of CHC in oenocytes, taking more than 10min. Quantitative modifications of some CHC may induce changes in self-perception and motivational states of the animals causing the increased latency to court females seen in our study, only 60min after fights. Our observation is consistent with others studies presenting evidence that defeated flies have a reduced motivation to court females [29] and to attempt copulation [27]. Besides, Filice and collaborators have revealed that subordinates have a mating duration slightly increased compared to dominants but the average number of offspring sired by both males were equivalent [29], suggesting that males' fitness is not related to the time invested in the mating part *per se*. However, in our study, we found that mating duration was not different between subordinates and dominants in a non-competitive context, while it was reduced in a competitive context. Such discrepancy could be explained by the different experimental protocols. Although different aggression and courtship assays protocols were used in these studies, the conclusion is consistent: previous defeat negatively affect courtship motivational state of males. Interestingly, multiple defeats have been shown to induce long-lasting loser effect that persist at least a day and dependent of *de novo* protein synthesis [11, 22]. In addition, Kim and collaborators have shown that multiple defeats also induce a reduction of courtship behaviours that can be observed after 1 day [22]. Put together, these results suggest that fighting experience can induce changes in males' internal state in both short- and long-term ways, depending on how many defeats animals have experienced.

In a competitive context where dominants and subordinates compete for a female, our findings align with the existing literature, confirming that prior fighting experience has an influence on reproductive success. However, in our study, the reproductive success of losers was significantly decreased only in a competitive context, suggesting a potential interaction between experiencing prior defeat and competing against an opponent for a female. This

hypothesis can be reinforced by the fact that losers tend to i) show lower reproductive success in a non-competitive context, and ii) spend less time displaying UWE toward the female in a competitive context. This interaction -fighting experience/presence of opponent- could also explain why, in this competitive context, losers have same likelihood of courting first the female. Competing against an opponent for females could overcome the negative effect of prior defeat, and increase losers' motivation to court females. However, as subordinates tend to perform fewer UWE in a competitive context, they might be ultimately less likely to mate with the female. While we cannot completely rule out the possibility that initial intrinsic differences, such as distinct energetic resources or courtship abilities, existed between actual losers and winners in our study, Teseo et al. in 2016 [27] provided supporting evidence against this hypothesis. They measured courtship behaviours before and after fighting experiences and demonstrated that decrease in courtship performance were observed only after losing experiences. This supports the idea that our observations are experience-dependent.

A question still remains: What serves as a predictor of males' reproductive success? Is it the males' fighting performances themselves or their subsequent dominance status that influence their reproductive success? In any scenario, previous fighting experiences may have crucial direct or indirect impacts on behavioural and phenotypical traits involved in courtship and reproductive success of dominant and subordinate males perceived by females. Distinguishing between the influence of males' fighting experiences and mating choice of females is a complex task. While some studies suggest that previous fighting experience may impact males' motivation to court females [17, 27–29], some others show that these are some males' phenotypic traits perceived by females that explain differences in reproductive success after fights [23, 41]. In another context, such as eavesdropping situation, female can assess dominance status of males by having watched their aggressive interactions. This effect has already been reported in fish [42] or birds [43]. Even though female flies are fitted with social learning capacities and can base their mating choice according to other females mating choice [44], to date, there have been no studies examining whether female flies after observing male fights, can accurately gauge the dominance status of males, and subsequently exhibit a mating preference for dominant males.

Development of winner and loser effects can be accompanied by changes in the quantity of chemical or olfactory signals emitted, leading to communication of males' dominance status to potential rivals and mates. This, in turn, can influence further social interactions. This has been shown in snapping shrimp and crayfish, that males can assess dominance status of opponent using olfactory cues [37], or in rodents, that female can detect odours of less aggressive males [45]. Results of our study indicate that, under our experimental conditions, females did not show an active preference for either decapitated dominants or subordinates based on chemicals, physicals or behavioural traits. This suggests that mating success may not primarily hinge on females themselves, or that females rely on additional behavioural and sensory cues not available when males are decapitated. If females do have preferences for winners, it would likely be influenced more by behavioural cues (such as UWE and song quality) rather than intrinsic traits alone (such as pheromones, size, etc.), or possibly a combination of both. To explore this hypothesis, one can perform experiment inspired from Loranger & Bertram's work [25] to see whether females would show an active preference towards alive males by blocking physical contact between dominants, subordinates and females.

Other studies have examined how fighting experience can modulate female fertility and number of offspring produced. In crickets, it has been shown that females that mated with dominant males laid more eggs [46], suggesting that females can either allocate more energy after mating with dominants to produce offspring, or that dominant males can enhance offspring sired by females. Fighting experiences can also affect females' fertility through various

mechanisms. *i*. Physiological effects: sperm quality of males can be altered by stress of aggressive interactions, potentially leading to fewer fertilized eggs after mating [47]. *ii*. Resource access: dominants would have better access to food, which can indirectly benefit females and their progeny [48] *iii*. Female choice: female preference for dominant may enhance female fertility by increasing the genetic quality of their offspring [49]. *iv*. Sperm competition: sperm of dominants may have competitive advantage in fertilizing eggs [50]. *v*. Territoriality: dominant males may create secure environments where females can raise their offspring more successfully, indirectly enhancing female fertility. *vi*. Males attractiveness: Females may be attracted to the displays and behaviours of males during fights, which can represent a signal of male quality, potentially increasing female interest and subsequently, fertility [51]. Collectively, these studies demonstrate how previous fighting experience significantly affect males' courtship behaviours, their reproductive success and their overall fitness, and might modulate female fertility. Nevertheless, it remains challenging to distinguish whether these impacts are direct or indirect, and whether they originate from males, females, or both.

## Conclusions

Fighting experiences may lead to deep changes in animals' behaviour and physiology. In general, consequences of previous defeats tend to be more profound, widespread and long-lasting than those of previous victories. Consistent with the existing literature, our findings in males *D. melanogaster* indicate that losing a fight leads to behavioural changes in subordinates: reduction in their territoriality, their motivation to court females, and their likelihood to mate in a competitive context. These effects can be observed at different temporalities, with an immediate impact on territorial defence, and after one hour on courtship and mating behaviours. However, fighting experiences have no influence either on males' ability to differentiate potential mates nor on females' choice to choose a mating partner. Overall, our findings contribute to a better understanding of how previous fighting experiences influence different behaviours that play a critical role in determining animals' fitness.

## Supporting information

**S1 Fig. The CVI extinction only differ in a competitive context.** (**A**) The CVI extinction do not differ between Winners, Losers, Naïve 10min (Chi$^2$ = 0.4 and p = 0.83) and (**B**) 60min (Chi$^2$ = 3.5 and p = 0.17) after fight in non-competitive context. (**C**) However, CVI does decline faster in losers in competitive context against winners.
(PDF)

**S1 File. Raw data collected from experiments.** Each sheet corresponds to a figure. Outliers detected by Grubb's test are in italic, blue and signalled with an *. Mean and Sd are calculated when it is possible.
(XLSX)

## Acknowledgments

We would like to thank members from the IVEP team at the Research Center for Animal Cognition for their comments on the manuscript. Especially, we would like to thank Anthony Defert (CRCA, Toulouse, France) and Anthony Roig (CNR, Instituo di Scienze e Tecnologie della Cognizione, Rome, Italy) who helped us with the statistical analysis, and thank the Edward Kravitz laboratory at Harvard Medical School (HMS, Boston, USA) for collecting preliminary results.

## Author Contributions

**Conceptualization:** Severine Trannoy.

**Data curation:** Antoine Prunier, Severine Trannoy.

**Formal analysis:** Antoine Prunier, Severine Trannoy.

**Funding acquisition:** Severine Trannoy.

**Project administration:** Severine Trannoy.

**Supervision:** Severine Trannoy.

**Writing – original draft:** Antoine Prunier.

**Writing – review & editing:** Severine Trannoy.

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
