## [Decision Letter · Decision Letter 0]

14 Jan 2024

PONE-D-23-38823Learning from fights: males' social dominance status impact reproductive success in <drosophila melanogaster="">.</drosophila>PLOS ONE

Dear Dr. Trannoy,

Thank you for submitting your manuscript to PLOS ONE. After careful consideration, we feel that it has merit but does not fully meet PLOS ONE’s publication criteria as it currently stands. Therefore, we invite you to submit a revised version of the manuscript that addresses the points raised during the review process.

We look forward to receiving your revised manuscript.

Kind regards,

Gregg Roman, PhD

Academic Editor

PLOS ONE

Journal Requirements:

"This work was supported by the French Research National Agency (ANR) (ANR-19-CE37-0018-01), and the Fondation Fyssen (190573) to S.T. Funders had no role in study design, data collection and analysis, decision to publish or preparation of the manuscript. We thank Anthony Defert who helped with the statistics. We also thank members from the IVEP team at the Research Center for Animal Cognition for their comments on the manuscript (CRCA, Toulouse, France) and from the Edward Kravitz laboratory at Harvard Medical School (HMS, Boston, USA) for collecting preliminary results."

"This work was supported by the French Research National Agency (ANR) (ANR-19-CE37-0018-01), and the Fondation Fyssen (190573) to S.T. Funders had no role in study design, data collection and analysis, decision to publish or preparation of the manuscript. 

https://anr.fr/

https://www.fondationfyssen.fr/en/

5. We note that your Data Availability Statement is currently as follows: All relevant data are within the manuscript and its Supporting Information files.

Additional Editor Comments (if provided):

Both reviewers found strong merit in the experiments and the manuscript.  The work is important an interesting - congratulations.  There were a few questions, however, that need to be addressed.  Reviewer 2 outlined  six major concerns, and I would like you to address all six prior to resubmission. Please also address the minor points as raised.  It is likely that responses to these comments many require new analyses and more refined commentary, but  not necessarily any new experiments. I anticipate that careful and thoughtful responses will greatly strengthen the paper and its impact.   In Major Concern (MC) #1, dealing with the meaning of the measures used to assess reproductive competency,  reviewer #2 brings up excellent questions regarding the interpretation of courtship assays that I believe can be handled through discussion of the points raised in the MC.   However, it may also be possible to add and experiment to make more clear what the effect is on motivation.   In MC #3, reviewer #2 also had questions about your statistical approach.   Reviewer #1 had similarly questioned the sensitivity of the non-parametric analysis.  I agree with Reviewer #2 that you should consider additional analyses for Fig 3B, but dependending on what you find, it may not require changing the Mann-Whitney test.    Several other MC (e.g., #s 4, 5 &6), all deal with the interpretations and conclusions.  These are all thoughtful comments, and I would like to see detailed responses to each.   

Reviewers' comments:

Reviewer's Responses to Questions

**Comments to the Author**

1. Is the manuscript technically sound, and do the data support the conclusions?

Reviewer #1: Yes

Reviewer #2: Partly

2. Has the statistical analysis been performed appropriately and rigorously? 

Reviewer #1: Yes

Reviewer #2: No

3. Have the authors made all data underlying the findings in their manuscript fully available?

Reviewer #1: No

Reviewer #2: No

4. Is the manuscript presented in an intelligible fashion and written in standard English?

Reviewer #1: Yes

Reviewer #2: Yes

5. Review Comments to the Author

Reviewer #1: This manuscript meets the PLOS ONE publication requirements as is. The data used to produce the figures was not made available for review. I am concerned with the use of non-parametric statistical tools to analyze behavioral assays, that given the sample sizes, may be best analyzed with parametric tools and suggest that the authors consider reanalyzing their data as parametric tools have greater statistical power. I suggest the authors have an English reader correct grammatical errors. Regardless, the manuscript meets the PLOS NE requirements as is.

Reviewer #2: The evolution of fighting behavior is often ascribed to its positive effect on fitness, but the number of studies about the effect of a result of competition on reproductive success has been relatively limited. In this background, the current study is valuable by performing experiments that behaviorally dissect the effect of losing a fight on mating behavior in Drosophila melanogaster. The method used in the behavioral assay is excellent. Although the results were presented adequately, I have some concerns in the way of interpretation of them.

Major concerns

1.

I wonder whether this study succeeded in showing that experience-dependent changes in the losers decreased their mating success. To evaluate the losers' intrinsic mating performance, the mating success rate in the non-competitive assay should be the most direct indicator (Fig.2E). Although I can see the trend where the losers were less successful than the winners (and the winners were less than naives, interestingly), the differences were not statistically supported, unfortunately. In contrast, mating success was significantly different between W and L in the competitive assay (Fig. 3A). A straightforward interpretation of these results would be "no experience-dependent effect but direct interaction between males resulted in the difference of mating success between them".---Comparing the courtship index (Fig. 2C vs Fig. 3B; lower in the competitive assay) or the mating duration (Fig. 2F vs Fig. 3F; shorter in the competitive assay), I suspect there was substantial interference from the opponent male in the competitive assay. And if the losers' courting motivation was reduced, we could expect that the courting 1st was also biased to the winners as well as the mating 1st (Fig.3A; in fact it seems slightly skewed to the winners from 50%).

Collectively, for me it looks a little tricky to say that the experience of losing a fight decreased the courtship motivation, resulting in less mating success in the competitive assay. Instead, it may be more simple to understand the results as that direct interaction was a major factor contributing to the fitness gain of the winners. In this context, no difference in the non-competitive assay can be employed to exclude the possibility of a self-sorting effect, by which winners were automatically biased to be more intrinsically energetic or motivated regardless of the fight outcome.

Alternatively, it may be possible to argue that a faint effect of the losing experience (which was marginally non-significant in the non-competitive assay) was exaggerated in the competitive assay and became visible. I agree that there were significant differences at least in some indicators (Fig. 1AB, 2A). Even though they didn't result in a significant difference in mating success in the non-competitive assay (Fig. 2E), they might have been enough to produce a substantial difference in the competitive assay. If authors think in this way, I would like to see a more explicit explanation of their thoughts. It is not very clear in the current manuscript in this regard. (e.g., Line 309--310)

2.

In Fig. 1A, I can see many individuals had a latency of zero seconds, which I suspect represent the flies staying on the food cup at the end of the 20 min fighting period. This is good proving that the experimental manipulation of inserting the partition was performed so gently that flies were not disturbed, as described in the Method section. On the other hand, it may not be good for the comparison between W and L, because more W were likely to be on the food cup at the end of the fighting period. They "stayed" rather than "returned" to the food cup, and including them in the statistical analysis should bias the result to support the difference between W and L. It is more conservative if the difference is still supported even when those "zero" incidents are excluded from the analysis.

3.

In Fig. 3B, the Mann-Whitney U test was used to examine the difference. It is not wrong to apply this test here, but it is less sensitive particularly when the ranges are overlapping to each other like in this case. Although the courtship index appears to be proportion data, practically they are time data with truncation (data points at 100% represent the truncated observation in which courtship might have continued further). In this sense, the courtship index has more affinity to longevity (survival) data with truncation, for which dedicated statistical methods have been developed. I feel it is too early to conclude that there was no difference in Fig. 3B. Suppose the observation had been much longer than 10 min. Courtship of many W males would have continued more than 10 min, whereas that of L males might have stopped earlier, making the difference more substantial. I feel sorry I can't specify the perfectly appropriate statistical method here (I may use GLM with binomial error distribution if I need to perform the same analysis), but strongly recommend considering other methods to analyze this data.

4.

Fig. 4CD: Can this be the test of the dominant males' status? Males were decapitated, therefore behavioral differences (reflecting social status) should not be presented to the females. I understand that the males needed to be decapitated to observe the purely female preference. But this treatment inevitably limits the interpretation of the result of this experiment. Instead, it may be allowed to state that male behavior is necessary to produce the difference in mating success. The related statements in the manuscript need to be tamed. For me, this experiment again seems testing the self-sorting effect, such as winners tend to smell good (and females prefer it) or be large (and females prefer larger males), etc.

5.

As I wrote above, differences observed in Fig. 3BC may reflect the interference by the opponent male (which may be more intense by W than L), rather than the experience effect. Statements (and thoughts) considering this possibility are preferable to appear in Results or Discussion.

6.

I would like to have the authors' thoughts on why the difference was obvious after 60 min but not in 10 min in Fig. 2A (I'm also curious about the same tendency in Fig. 2CE).

Minor points

7.

The term "territory" seems to be used at least in three different meanings: the arena (e.g., Line 144), the food cup (Line 157), and the true conceptual territory (Line 160). By its definition, territory is rather a hard concept to be proven experimentally, and the term should be used carefully. I recommend not to use "territory" where it is not strictly supported, particularly in Fig.1 and related sentences in Results (i.e., "return on the territory" or "time spent on the territory": they should be substituted with "the food cup").

8.

Related to the experiment shown in Fig. 4AB, the method of decapitation of males after the fighting period (how to retrieve them, etc.) seems not described.

9.

The number of replicates is preferable to be shown in Fig.2E and 3A because it influences the confidence interval.

10.

Line 173: insert "(UWE)"?

11.

Line 359: "induce a decreased" ??

6. PLOS authors have the option to publish the peer review history of their article (what does this mean?). If published, this will include your full peer review and any attached files.

Reviewer #1: No

Reviewer #2: No

---

## [Author Response · Author response to Decision Letter 0]

2 Feb 2024

Reviewers’ comments:

Reviewer #1: 

This manuscript meets the PLOS ONE publication requirements as is. The data used to produce the figures was not made available for review.

###We furnished all the raw data as a new Supporting Information excel files.

I am concerned with the use of non-parametric statistical tools to analyze behavioral assays, that given the sample sizes, may be best analyzed with parametric tools and suggest that the authors consider reanalyzing their data as parametric tools have greater statistical power. 

###While it is true that parametric tests offer greater statistical power, data from behavioral assays often deviate from a standard normal distribution. Consequently, researchers in the field tend to consistently opt for non-parametric tests.

Nevertheless, we subjected our entire dataset for winners and losers to the Shapiro-Wilk test for normality. The results indicated that the majority of our dataset did not pass the Shapiro-Wilk test, suggesting a departure from a standard normal distribution. Hence, the use of non-parametric tests, such as the Mann-Whitney test, is recommended.

For instance, in Fig 1B, the distribution for both winners and losers failed the Shapiro-Wilk test. Similarly, in Fig 1C, the distribution for winners did not meet the normality criterion, while the distribution for losers passed the test. In such cases, it is advisable to employ non-parametric tests.

Despite these findings, we performed a comparison of the results obtained from non-parametric and parametric tests for our Figure 1 dataset. The results demonstrated that both types of tests led to the same interpretation of the results. For this reason, we have decided to consistently apply non-parametric tests throughout the entire manuscript.

- Fig 1B Latency to return to the food cup

o Non-parametric Mann-Whitney: U=69, p=0.1038

o Parametric t-test: t=1,843, df=28, p=0.076

- Fig 1C: Time spent on food cup

o Non-parametric Mann-Whitney: U=92, p=0,0003

o Parametric t-test: t=3.902, df=42, p=0.0003

- Fig 1D: Midline crossings

o 10min Non-parametric Mann-Whitney: U=34.5 and p=0.6486 

o 10min parametric t-test: t=0.5682, df=16, p=0.5778. 

o 60min Non-parametric: U=112, p=0.9919

o 60min Parametric t-test: t=0.1024, df=28, p=0.9192

I suggest the authors have an English reader correct grammatical errors. Regardless, the manuscript meets the PLOS NE requirements as is.

###Thank you. We hope we succeed to correct grammatical errors in our revised version.

Reviewer #2: 

The evolution of fighting behavior is often ascribed to its positive effect on fitness, but the number of studies about the effect of a result of competition on reproductive success has been relatively limited. In this background, the current study is valuable by performing experiments that behaviorally dissect the effect of losing a fight on mating behavior in Drosophila melanogaster. The method used in the behavioral assay is excellent. Although the results were presented adequately, I have some concerns in the way of interpretation of them.

MC#1: I wonder whether this study succeeded in showing that experience-dependent changes in the losers decreased their mating success. To evaluate the losers' intrinsic mating performance, the mating success rate in the non-competitive assay should be the most direct indicator (Fig.2E). Although I can see the trend where the losers were less successful than the winners (and the winners were less than naive, interestingly), the differences were not statistically supported, unfortunately. In contrast, mating success was significantly different between W and L in the competitive assay (Fig. 3A). A straightforward interpretation of these results would be "no experience-dependent effect but direct interaction between males resulted in the difference of mating success between them». ---Comparing the courtship index (Fig. 2C vs Fig. 3B; lower in the competitive assay) or the mating duration (Fig. 2F vs Fig. 3F; shorter in the competitive assay), I suspect there was substantial interference from the opponent male in the competitive assay. And if the losers' courting motivation was reduced, we could expect that the courting 1st was also biased to the winners as well as the mating 1st (Fig.3A; in fact it seems slightly skewed to the winners from 50%).

Collectively, for me it looks a little tricky to say that the experience of losing a fight decreased the courtship motivation, resulting in less mating success in the competitive assay. Instead, it may be more simple to understand the results as that direct interaction was a major factor contributing to the fitness gain of the winners. In this context, no difference in the non-competitive assay can be employed to exclude the possibility of a self-sorting effect, by which winners were automatically biased to be more intrinsically energetic or motivated regardless of the fight outcome.

Alternatively, it may be possible to argue that a faint effect of the losing experience (which was marginally non-significant in the non-competitive assay) was exaggerated in the competitive assay and became visible. I agree that there were significant differences at least in some indicators (Fig. 1AB, 2A). Even though they didn't result in a significant difference in mating success in the non-competitive assay (Fig. 2E), they might have been enough to produce a substantial difference in the competitive assay. If authors think in this way, I would like to see a more explicit explanation of their thoughts. It is not very clear in the current manuscript in this regard. (e.g., Line 309--310).

###MC#1 raised an important point. We modified the text in our manuscript and added a paragraph in the discussion section to support the hypothesis that there is a potential interaction between experiencing a defeat and being in competition against another male to access females, which would reinforce the losers’ motivation to court female in a competitive context. We hope our modifications answer this point.

MC#2: In Fig. 1A, I can see many individuals had a latency of zero seconds, which I suspect represent the flies staying on the food cup at the end of the 20 min fighting period. This is good proving that the experimental manipulation of inserting the partition was performed so gently that flies were not disturbed, as described in the Method section. On the other hand, it may not be good for the comparison between W and L, because more W were likely to be on the food cup at the end of the fighting period. They "stayed" rather than "returned" to the food cup, and including them in the statistical analysis should bias the result to support the difference between W and L. It is more conservative if the difference is still supported even when those "zero" incidents are excluded from the analysis. 

###The point raised is very interesting. We decided to take into account this comment by modifying the Figure 1. We added a new Fig1A that represents the number of winners and losers that stayed on the food cup between the end of the fight and the start of the territory assays. We also statistically analysed these proportions and concluded that significantly more winners stayed on the food cup than losers (Chi²=4.2712, df=1 p=0.0388). Also, we decided to exclude these flies (9 winners and 3 losers) from the “latency to return to food cup” which is now the Fig 1B. We stated this point in the material and method section. However, the difference between losers and winners is no more significant, we only observe a tendency for losers to take more time returning to the food cup (Mann-Whitney test, U=69, p=0.1038). We updated all the statistics. We hope our modifications would answer the point raised in MC#2.

MC#3: In Fig. 3B, the Mann-Whitney U test was used to examine the difference. It is not wrong to apply this test here, but it is less sensitive particularly when the ranges are overlapping to each other like in this case. Although the courtship index appears to be proportion data, practically they are time data with truncation (data points at 100% represent the truncated observation in which courtship might have continued further). In this sense, the courtship index has more affinity to longevity (survival) data with truncation, for which dedicated statistical methods have been developed. I feel it is too early to conclude that there was no difference in Fig. 3B. Suppose the observation had been much longer than 10 min. Courtship of many W males would have continued more than 10 min, whereas that of L males might have stopped earlier, making the difference more substantial. I feel sorry I can't specify the perfectly appropriate statistical method here (I may use GLM with binomial error distribution if I need to perform the same analysis), but strongly recommend considering other methods to analyze this data. 

###Thank you for this comment. You proposed to considered the CVI data more as survival data with truncation instead of proportion. In the literature, articles using CVI as a measure compare it as proportion and often use Kruskal-Wallis test (e.g. see Krstic, Boll & Noll, 2013). Yet, your remark is very pertinent and we performed survival analysis based on all our CVI data. The results are available as Supplementary Figures. Briefly, these analyses did not show any difference in non-competitive context at 10min (Chi²=0.4, Df=2, p=0.8) and 60min (Chi²=3.5, Df=2, p=0.2). However, as you suspected, we found statistical difference in a competitive context. Losers extinguished faster their courtship parade than winners (Chi²=4.3, Df=1, p=0.037). We therefore decided to kept the analysis with non-parametric tests in the main figures, and we added a Supplementary file concerned with the survival analysis.

MC#4: Fig. 4CD: Can this be the test of the dominant males' status? Males were decapitated, therefore behavioral differences (reflecting social status) should not be presented to the females. I understand that the males needed to be decapitated to observe the purely female preference. But this treatment inevitably limits the interpretation of the result of this experiment. Instead, it may be allowed to state that male behavior is necessary to produce the difference in mating success. The related statements in the manuscript need to be tamed. For me, this experiment again seems testing the self-sorting effect, such as winners tend to smell good (and females prefer it) or be large (and females prefer larger males), etc. 

###In the MC#4, you pointed out that our test of female preference for decapitated winner and loser males cannot fully test the dominant males’ status, because most of the cues signalling this status cannot be displayed through behaviours. We totally agree with this point, and we already discussed it on the first version of our manuscript « Results of our study indicate that, under our experimental conditions, females did not show an active preference for either decapitated dominants or subordinates. This suggests that mating choice may not primarily hinge on females themselves, or that females rely on additional behavioural and sensory cues not available when males are decapitated. ». To answer your point, we developed our thoughts to make it clearer in the discussion. Regarding your remark about testing the self-sorting effect instead of experience, it may be partly true. We did our best to restraint to the minimum the size difference between the males. However, we could not control the cuticular hydrocarbon (CHC) profile before and after the fight. Since the females did not show any preference toward decapitated winner or loser, our guess would be that males did not differ from their cuticular profile, or at least not enough to induce females’ preferences. We therefore elaborated our discussion to answer this point.

MC#5: As I wrote above, differences observed in Fig. 3BC may reflect the interference by the opponent male (which may be more intense by W than L), rather than the experience effect. Statements (and thoughts) considering this possibility are preferable to appear in Results or Discussion. 

###We agree, we added a paragraph in the discussion section to elaborate on this point. 

MC#6: I would like to have the authors' thoughts on why the difference was obvious after 60 min but not in 10 min in Fig. 2A (I'm also curious about the same tendency in Fig. 2CE). 

###That is a very good question. We added a reflexion about this in the discussion section. In few words, we think that synthesis of CHC may be involved. When a fly is experiencing a defeat or victory, it may trigger new synthesis of specific CHC to potentially convey its social status. Quantitative CHC changes may influence the perception of this fly by conspecifics, or even its own perception, and in turn induce behavioural changes or changes in the motivational state of the fly. New CHC synthesis requires some time - we expect more than 10min -, which can explain why difference can only be found 60 minutes after the fight.

Another hypothesis would be that fights induce physiological changes with different kinetic. 10min may be too soon after the fight for these changes to be processed. 

Minor points

7. The term "territory" seems to be used at least in three different meanings: the arena (e.g., Line 144), the food cup (Line 157), and the true conceptual territory (Line 160). By its definition, territory is rather a hard concept to be proven experimentally, and the term should be used carefully. I recommend not to use "territory" where it is not strictly supported, particularly in Fig.1 and related sentences in Results (i.e., "return on the territory" or "time spent on the territory": they should be substituted with "the food cup").

###We agreed with this point, and changed the text accordingly. We changed some terms “territory” and replaced them by food cup as suggested. Some of them were kept in the text, since in we consider here the territory as the food cup, edges included.

8. Related to the experiment shown in Fig. 4AB, the method of decapitation of males after the fighting period (how to retrieve them, etc.) seems not described.

###This was an oversight on our part. We added the methodology used for decapitating flies in our assays.

9. The number of replicates is preferable to be shown in Fig.2E and 3A because it influences the confidence interval.

###The numbers of replicates were added in the legends of figures 2E and 3A.

10. Line 173: insert "(UWE)"?

###Done

11. Line 359: "induce a decreased" ??

###We replaced “induce a decreased" by “induce a reduction"

Kind regards,

Antoine Prunier & Séverine Trannoy.

---

## [Decision Letter · Decision Letter 1]

19 Feb 2024

Learning from fights: males' social dominance status impact reproductive success in <drosophila melanogaster="">.

PONE-D-23-38823R1</drosophila>

Dear Dr. Trannoy,

We’re pleased to inform you that your manuscript has been judged scientifically suitable for publication and will be formally accepted for publication once it meets all outstanding technical requirements.

Kind regards,

Gregg Roman, PhD

Academic Editor

PLOS ONE

Additional Editor Comments (optional):

Reviewers' comments:

Reviewer's Responses to Questions

**Comments to the Author**

1. If the authors have adequately addressed your comments raised in a previous round of review and you feel that this manuscript is now acceptable for publication, you may indicate that here to bypass the “Comments to the Author” section, enter your conflict of interest statement in the “Confidential to Editor” section, and submit your "Accept" recommendation.

Reviewer #1: All comments have been addressed

Reviewer #2: All comments have been addressed

2. Is the manuscript technically sound, and do the data support the conclusions?

Reviewer #1: Partly

Reviewer #2: Yes

3. Has the statistical analysis been performed appropriately and rigorously? 

Reviewer #1: Yes

Reviewer #2: Yes

4. Have the authors made all data underlying the findings in their manuscript fully available?

Reviewer #1: Yes

Reviewer #2: Yes

5. Is the manuscript presented in an intelligible fashion and written in standard English?

Reviewer #1: Yes

Reviewer #2: Yes

6. Review Comments to the Author

Reviewer #1: (No Response)

Reviewer #2: (No Response)

7. PLOS authors have the option to publish the peer review history of their article (what does this mean?). If published, this will include your full peer review and any attached files.

Reviewer #1: **Yes: **Miguel de la Flor

Reviewer #2: No

---

## [Editor Report · Acceptance letter]

27 Feb 2024

PONE-D-23-38823R1 

PLOS ONE

Dear Dr. Trannoy, 

I'm pleased to inform you that your manuscript has been deemed suitable for publication in PLOS ONE. Congratulations! Your manuscript is now being handed over to our production team.

Kind regards, 

on behalf of

Dr Gregg Roman 

Academic Editor

PLOS ONE